# Impact of HIV-1 Infection on the Natural Progress of an Anti-HCV Positive Population in an Impoverished Village in China from 2009 to 2017

**DOI:** 10.3390/v14081621

**Published:** 2022-07-26

**Authors:** Xinjie Li, Yuantao Li, Yuqi Zhang, Yue Yin, Jing Tu, Qiang Xu, Hua Liang, Tao Shen

**Affiliations:** 1Department of Microbiology and Infectious Disease Center, School of Basic Medical Sciences, Peking University, Beijing 100191, China; xinjieli@hsc.pku.edu.cn (X.L.); zhangyuqi_@bjmu.edu.cn (Y.Z.); yuey@bjmu.edu.cn (Y.Y.); tujing@hsc.pku.edu.cn (J.T.); xuqiang@hsc.pku.edu.cn (Q.X.); 2China Sinopharm International Corporation, Beijing 100029, China; liyuantao@sinopharm.com; 3State Key Laboratory of Infectious Disease Prevention and Control (SKLID), National Center for AIDS/STD Control and Prevention, China CDC, Collaborative Innovation Center for Diagnosis and Treatment of Infectious Diseases, Beijing 102206, China

**Keywords:** hepatitis C virus (HCV), human immunodeficiency virus (HIV), liver fibrosis, aspartate aminotransferase to platelet ratio index (APRI), fibrosis 4 score (FIB-4)

## Abstract

Our study aimed to determine the impact of HIV coinfection on the natural progression of liver disease in treatment-naive HCV-infected patients. From 2009 to 2017, we tracked non-invasive markers of liver fibrosis and end-stage liver disease (ESLD)-associated mortality among HCV mono-infected and HIV/HCV coinfected patients in an impoverished village in China. The study cohort consisted of 355 HBsAg-negative and anti-HCV (+) or anti-HIV (+) patients recruited in July 2009, 164 of whom were diagnosed with HIV-1 infection. The surviving patients were re-evaluated in August 2017. During the follow-up, the disease status, liver biochemical, and non-invasive indicators of liver fibrosis (APRI and FIB-4) were measured. The transaminases ALT and AST were significantly higher in HIV-positive HCV resolvers (HIV+ HCVr) than in HIV-negative HCV resolvers (HCVr) (*p* = 0.019 and *p* < 0.0001, respectively). APRI and FIB-4 scores of HIV-positive chronic HCV carriers (HIV+ HCVc) were significantly higher than in HIV-negative chronic HCV carriers (HCVc) (*p* < 0.001). Similarly, APRI and FIB-4 scores were higher in the HIV+ HCVr group than in the HCVr group (*p_s_* < 0.001). From 2009 to 2017, the levels of ALT (*p* = 0.006), AST (*p* = 0.003), APRI (*p* = 0.015), and FIB-4 (*p* = 0.025) were significantly elevated in the HIV/HCV coinfected patients with CD4+ T counts below 500 cells/l. ESLD-related mortality was significantly greater in HIV/HCV-infected cases than in HCV mono-infected patients (73.3% vs. 31.3%, *p* = 0.009) among patients (*n* = 45) who died between 2009 and 2017 during follow-up. These findings suggest a higher risk of ESLD-related death and rapid progression of liver fibrosis in HIV/HCV coinfected individuals compared with HCV mono-infected patients. During HIV/HCV coinfection, HIV infection may aggravate HCV-associated liver injury.

## 1. Introduction

Current evidence suggests that more than 70 million individuals globally have been infected with human immunodeficiency virus (HIV). Given that HIV and hepatitis C virus (HCV) share the same transmission mode, around 30% of HIV-positive patients are also infected with HCV, a major cause of chronic viral hepatitis, cirrhosis, and hepatocellular cancer [1]. Following the widespread use of highly active antiretroviral therapy (HAART), acquired immune deficiency syndrome (AIDS)-related mortality has decreased dramatically in recent years. In contrast, HCV-related liver disease has become the leading cause of death among AIDS patients, with nearly 40% to 50% of HIV/HCV coinfected patients dying from end-stage liver diseases (ESLD) [2]. Hepatic fibrosis is a reversible, dynamic response to the liver injury, induced by chemical, metabolic, or viral damage, and results from an imbalance in extracellular matrix (ECM) protein turnover, characterized by accelerated synthesis and reduced degradation. During chronic HCV infection, hepatic injury is prolonged with persistent inflammation, and ECM protein accumulation exceeds their degradation [3], triggering fibrogenesis [4] and leading to a progressive substitution of liver parenchyma by scar tissue. This phenomenon results in regenerating hepatocyte nodules, a feature that determines fibrosis progression to cirrhosis [5].

Numerous cohort studies have explored whether HIV coinfection exacerbates the course of HCV liver fibrosis, but the findings remain controversial [6]. Avihingsanon A et al. [7] and Li Vecchi et al. [8] reported a higher incidence of disease progression and advanced fibrosis in coinfected individuals than in HCV mono-infected individuals. Mazzocato S. et al. [9] and Tovo CV et al. [10], on the other hand, documented similar rates of disease progression in individuals with HCV mono-infection and HIV/HCV coinfection. Notably, in most studies, several important factors, including alcohol use, smoking, gender, and length of HIV infection and treatment modality, were unevenly distributed across the HIV/HCV coinfection cohort and the HCV group, potentially biasing the statistical results.

In this study, we aimed to longitudinally and cross-sectionally investigate the effect of HIV coinfection on the progression of HCV fibrosis. The survey was conducted in a consistent background cohort, which minimizes the interference of non-disease factors. During the follow-up period, no HCV-related medication was administered to any of the patients in this cohort, Therefore, the impact of HIV coinfection on the natural progression of illness in HCV carriers could be observed.

## 2. Materials and Methods

### 2.1. Study Population

The initial screening phase of this study was conducted in July 2009, and 355 HBsAg-negative, anti-HCV (+), or anti-HIV (+) individuals from an impoverished village in central China were recruited. More than 90% of the participants were former plasma donors (FPD) who had a nonstandard history of paid blood donation in the 1990s, and the remainder were their parents, spouses, or children. As illustrated in Figure 1, the study participants (*n* = 355) consisted of HCV-positive patients (HCV RNA+, *n* = 227) and HCV-negative patients (HCV RNA-, *n* = 128), based on HCV RNA test results. Regarding anti-HCV and anti-HIV status, HCV-positive patients were subdivided into HIV-negative chronic HCV carriers (HCVc, *n* = 129) and HIV-positive chronic HCV carriers (HIV+ HCVc, *n* = 98). HCV-negative patients included HIV-negative HCV resolvers (HIVr, *n* = 62), HIV-positive HCV resolvers (HIV+ HCVr, *n* = 48), and HIV mono-infected patients (HIV+, *n* = 18).

Using a standardized questionnaire, all participants were interviewed, and data were recorded by trained medical staff regarding their demographic information, underlying diseases, blood donation histories, and use of antiviral or antiretroviral medications. The survey results indicated that none of the participants had received any form of HCV-specific antiretroviral therapy, while all HIV-positive subjects received regular or intermittent first-line highly active antiretroviral therapy (HAART). The HAART regimen usually consists of two nucleoside reverse transcriptase inhibitors (NRTI): either azidovudine plus didanosine or stavudine plus lamivudine, and nevirapine, a non-nucleoside reverse transcriptase inhibitor (NNRTI). Table 1 summarizes the demographic characteristics and test results for all patients enrolled in July 2009. A follow-up investigation was conducted in August 2017 again, and 206 patients were successfully contacted, while 104 were lost to follow-up, and 45 died. In addition, 16 out of 81 HCV-positive patients and 9 out of 53 HIV+ HCV-positive patients were cured after receiving HCV-targeted therapy during follow up. Furthermore, two of the HCVr patients (*n* = 38) were re-infected with HCV and two new HCV infections were found in the HIV+ group (*n* = 9). The patients mentioned above were excluded from the cohort to ensure the accuracy of the study results. Therefore, during the follow-up conducted in July 2017, the final enrollment included HIV- HCVc (*n* = 65), HIV+ HCVc (*n* = 44), HCVr (*n* = 36), HIV+ HCVr (*n* = 25), and HIV+ mono-infected (*n* = 7) patients for analysis. The flow chart of subjects for this study is shown in Figure 1.

The study was approved by the institutional review authorities of Peking University Health Science Center (Approval ID: PKUPHLL20090011). All patients provided written informed consent before enrollment in the study.

### 2.2. HIV and HCV Seropositive Screening and Confirmation

The plasma anti-HCV antibody was detected by Abbott Architect anti-HCV assays (Abbott GmbH & Co. KG, Wiesbaden, Germany) and confirmed by HCV-RIBA assays (Wantai Biological Pharmacy, Beijing, China). The plasma anti-HIV antibody was screened by ELISA (GBI Biotech Co., Ltd., Beijing, China) and confirmed by HIV Blot 2.2 WB assays (Genelabs Diagnostics, Singapore).

### 2.3. Quantification of HCV-RNA, HIV-RNA, and CD4+/ CD8+ T-Cell Counts

The quantitation of plasma HCV RNA was performed using the Abbott Real-Time HCV Amplification Kit (Abbott Molecular Inc. Des Plaines, IL, USA). The plasma concentration of HIV-1 RNA was measured by the Standard Amplicor HIV Monitor assay, version 2.0 (Roche Diagnostics, Indianapolis, IN, USA). CD4+ T cell counts were measured by EDTA-treated whole blood for CD3/CD4/CD8/CD45 in TruCount tubes and analyzied with a FACSCalibur flow cytometer (Becton Dickinson, San Jose, CA, USA). The absolute numbers of CD4+ T lymphocytes were determined using MultiSET software (BD Bioscience, San Jose, CA, USA).

### 2.4. Evaluation of Liver Fibrosis

Liver fibrosis was evaluated using the aspartate aminotransferase to platelet ratio index (APRI) and the FIB-4 fibrosis index. APRI was calculated according to the following formula: AST [IU/L]/(upper limit of normal range) × 100/platelet count (10^9^/L). The upper limit of the normal range of AST in this study was 40 IU/L. The FIB-4 index was calculated using a formula that included patient age, serum aspartate aminotransferase (AST) and alanine aminotransferase (ALT) concentrations, and platelet counts; FIB-4 index = (age [years] × AST [IU/L])/(platelet counts [10^9^/L] × ALT [IU/L]^1/2^).

### 2.5. Statistical Analyses

Wilcoxon matched pair test, Mann–Whitney U-test, and Pearson chi-square tests were performed using GraphPad Prism 9.0 software (Inc., San Diego, CA, USA) when necessary. All tests were two-tailed, and *p*-values < 0.05 were considered statistically significant.

## 3. Results

### 3.1. The HIV/HCV Coinfection Group Had Higher Transaminase Levels than the HCV Mono-Infection Group

ALT and AST are well-established as two key indicators for evaluating liver function. We first examined ALT and AST levels among follow-up patients with various HCV and/or HIV infection groups in 2009. No significant differences in ALT or AST were found between chronically HCV-infected patients (HCVc) and HIV/HCV coinfected patients (HIV + HCVc). In contrast, among HCV-resolved individuals, ALT (Figure 2A, *p* = 0.019) and AST (Figure 2B, *p* < 0.001) levels were significantly higher in HIV-positive patients (HIV + HCVr) than in HIV-negative patients (HCVr). During the follow-up in 2017, ALT and AST were reassessed in 109 HCV treatment naïve patients (HCVc, *n* = 65 and HIV + HCVc, *n* = 44). A longitudinal comparison between 2009 and 2017 showed that ALT levels of patients with normal values (<40 U/L) in 2009 experienced a substantial increase in 2017, either for the HCVc group or HIV + HCVc patients (*p_s_* < 0.001) (Figure 2C). A similar trend was observed in 2017 among subjects with normal AST values (<40 U/L) in 2009 (HCVc group: *p* = 0.005; HIV + HCVc group: *p* < 0.001) (Figure 2D).

### 3.2. The HIV/HCV Coinfection Group Had Higher APRI and FIB-4 Levels than the HCV Mono-Infection Group

APRI and FIB-4 are two crucial noninvasive indicators to track the progression of liver fibrosis. Patients with chronic HCV infection may develop liver fibrosis, leading to cirrhosis and potentially hepatocellular carcinoma. APRI and FIB-4 scores were calculated to determine the impact of coinfection with HIV on liver fibrosis. In 2009, both APRI (Figure 3A, *p* < 0.001) and FIB-4 (Figure 3B, *p* < 0.001) scores were significantly higher in the HIV+ HCVc group than in the HCVc group. Moreover, among spontaneous HCV resolved subjects, significantly higher APRI (Figure 3A, *p* < 0.001) and FIB-4 (Figure 3B, *p* = 0.001) levels were found in HIV-positive patients (HIV+ HCVr) than in HIV-negative subjects (HCVr). In accordance with clinical guidelines, an APRI score below 0.5 or FIB-4 less than 1.45 indicates non-fibrosis. Based on this, we examined APRI and FIB-4 level changes from 2009 to 2017 in HCV mono-infected and HIV/HCV coinfected individuals. It was found that APRI levels of HCVc patients with fairly normal scores (<0.5) in 2009 significantly increased in 2017 (*p* < 0.001), and the same was found in HIV/HCV coinfections (*p* = 0.001) (Figure 3C). Additionally, patients with normal FIB-4 values also exhibited elevated levels after 8 years (HCVc group: *p* < 0.001; HIV+ HCVc group: *p* = 0.007) (Figure 3D).

### 3.3. HIV Disease Progression Contributes to HCV Liver Fibrosis

Given that CD4+ T-cell counts ≥500 cells/μL were associated with improved HIV clinical prognosis [11], all HIV-infected patients enrolled in 2009 were separated into two groups based on CD4+ T-cell count: CD4+ T-cell ≥ 500 cells/μL group (*n* = 18) and CD4+ T-cell < 500 cells/μL group (*n* = 25). To investigate the influence of HIV progression on HCV liver fibrosis, the values of AST, ALT, APRI, and FIB-4 were analyzed and compared between distinct CD4+ T cell subgroups between 2009 and 2017. As demonstrated in Figure 4, all parameters increased considerably from 2009 to 2017 in the CD4+ T-cell count < 500 cells/L group (ALT: *p* = 0.006, AST: *p* = 0.003, APRI: *p* = 0.015, FIB-4: *p* = 0.025), but no significant changes were identified in the CD4+ T-cell count ≥ 500 cells/L group (Figure 4).

### 3.4. HIV/HCV Coinfection Was Associated with Higher Mortality in ESLD

Among the 227 chronic HCV patients enrolled in 2009, 34 died by 2017, including patients with HCV mono-infection (HCVc, *n* = 16) and HIV/HCV coinfection (HIV + HCVc, *n* = 18). Despite the HCVc group (12.4%, 16/129) and the HIV + HCVc group (18.4%, 18/98) having comparable overall mortality rates, it was noteworthy that ESLD was a significant contributor to death, accounting for 73.3% (11/15) of non-AIDS-related deaths in HIV/HCV coinfected cases, and 31.3% (5/16) in HCV mono-infected patients (*p* = 0.009) (Table 2). The significantly higher proportion of ESLD-related deaths among HIV/HCV coinfected patients suggests that HIV infection could exacerbate HCV progression, especially liver fibrosis.

## 4. Discussion

This study is based on a long-term follow-up of a cohort of 355 HBeAg-negative, anti-HCV, or anti-HIV-positive individuals whose initial infection was caused by nonstandard paid blood donation, without a history of drug abuse, or men who have sex with men (MSM). All individuals came from the same village in central China, where similar living conditions minimized the interference of non-disease-related variables. Several small-scale studies have indicated that APRI and FIB-4 scores are highly correlated with severe fibrosis in patients with chronic hepatitis B [12,13], consistent with the present study findings. The cross-sectional survey conducted in 2009 demonstrated that typical liver function indicators and overall fibrosis levels were significantly higher in HIV/HCV coinfected patients than in HCV mono-infected patients. Moreover, the pairwise comparison between 2017 and 2009 revealed that the rate of death associated with ESLD was significantly greater in HIV coinfected individuals during long-term follow-up, despite both HIV/HCV coinfected and HCV mono-infected individuals presenting significant liver function impairment and fibrosis progression after 8 years. These findings also validate that HIV coinfection might worsen HCV-associated liver fibrosis.

The above results may be attributed to the following events: (1) Participants in the cohort study were mainly infected with HIV in the late 1990s, and the free treatment policy of HARRT for HIV was not implemented until after 2004, with a 5–10-year treatment gap period. Therefore, the 2009 cross-sectional survey was primarily a comprehensive reflection of HCV disease progression among patients with HIV infection who were not receiving standardized treatment. (2) During 2009 and 2017, all HIV patients underwent long-term standardized antiviral therapy; thus, HIV coinfected individuals with favorable immune and recovery exhibited comparable liver injury and progression to HCV mono-infected individuals. (3) HIV coinfected individuals with severe immune dysfunction or poor recovery from antiviral treatment had a higher risk of death from end-stage liver diseases.

HIV infection causes immunosuppression, typically manifested by a decline in CD4 cell counts [14]. Herein, we found that transaminase levels increased more rapidly in HIV coinfected patients with low CD4 counts (<500 cells/μL) than in HCV mono-infected patients. The EASL (European Association for the Study of the Liver) [15] and AASLD [16] guidelines urge early initiation of highly active antiretroviral treatment regardless of the CD4 count. Although individuals in our cohort underwent HARRT therapy, they could not attain a CD4 count comparable to HIV-negative individuals, consistent with Aiuti F et al.’s review [17].

Over the years, combination antiretroviral therapy (cART) has dramatically reduced overall morbidity and mortality in HIV-infected populations. However, it should be borne in mind that other comorbidities could influence the clinical trajectory of HIV-infected patients as they live longer. Current evidence suggests an increased prevalence of liver-related complications in patients with HIV/AIDS, accounting for 9% of fatalities among unselected HIV-positives [18]. In our study, ESLD-related mortality was significantly greater in HIV/HCV coinfected patients, suggesting that HIV coinfected patients require careful monitoring and control of their liver damage profile and fibrosis development. The impact and immunological involvement of HIV in HCV infection remain poorly understood. Previous evidence indicated that HIV coinfected patients progress more rapidly toward liver fibrosis, leading to an increased incidence of liver complications. Several factors, including HIV viremia [19], CD4 lymphopenia [14], bacterial translocation [20,21], and the toxicity of some antiretroviral medications [22], are widely believed to play a role in the fibrotic process in HIV/HCV coinfected patients, which may act synergistically with traditional factors such as alcohol consumption, smoking, and drug abuse. All of these variables contribute to the course of fibrosis and catalyze the development of liver complications when combined.

There were limitations in our research. First, as the study participants came from a remote rural area, laboratory data collection and testing conditions were extremely limited, resulting in a lack of comprehensive data. Moreover, the limitations of follow-up studies affected the study findings to a certain extent owing to the loss of patients to follow-up or death several years after infection. In addition, the hepatotoxicity induced by antiretroviral medication itself is noteworthy, despite the fact that most patients with HIV mono-infection remained in a state of normalcy with no significant rise in liver function indicators during the long-term follow-up, which indicates to some extent that the hepatotoxic effects induced by antiretroviral therapy in this study were comparatively limited. However, it is possible that antiretroviral-induced hepatotoxicity was present and acted as a confounding factor during the first follow-up in 2009 or earlier. Last, all HIV samples derived from patients with blood-borne illnesses were of a relatively small sample size. It is widely acknowledged that the relationship between HIV and HCV infection is intricate. Indeed, studies on different study populations with heterogeneous infection pathways and lifestyles are likely to yield inconsistent findings, warranting the need for cross-validation in a larger population.

## 5. Conclusions

In conclusion, our longitudinal and cross-sectional analyses showed that HIV coinfection accelerates the natural progression of HCV liver fibrosis, which could guide treatment strategies or regimens for patients with HIV/HCV coinfection.

## Figures and Tables

**Figure 1 viruses-14-01621-f001:**
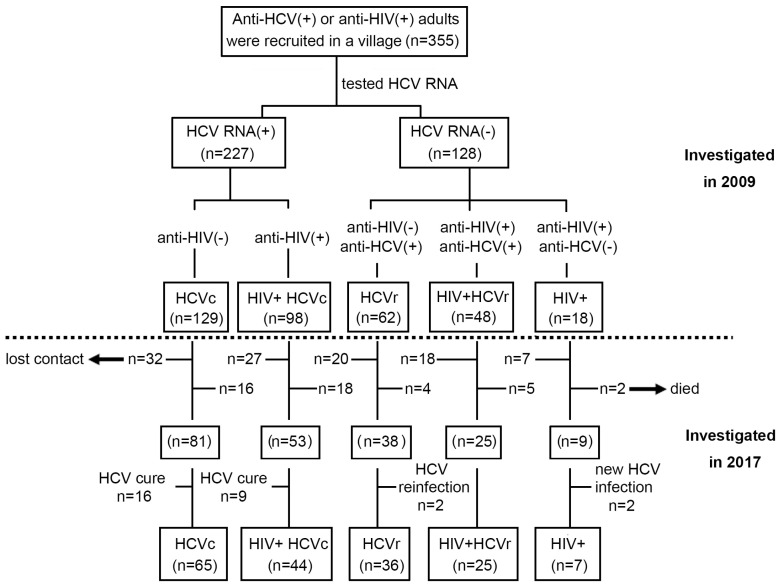
A flow diagram representing the recruitment of 355 anti-HCV (+) or anti-HIV (+) patients for the study. Above and below the dashed line are the initial follow-up visit in 2009 and the second visit in 2017, respectively. Patients with HIV − HCVc (*n* = 65), HIV+ HCVc (*n* = 44), HCVr (*n* = 36), HIV+ HCVr (*n* = 25), and HIV + mono-infection (*n* = 7) were enrolled for analysis after inclusion and exclusion. HCV, hepatitis C virus; HCVc, chronic HCV carriers; HCVr, HCV resolvers; HIV, human immunodeficiency virus.

**Figure 2 viruses-14-01621-f002:**
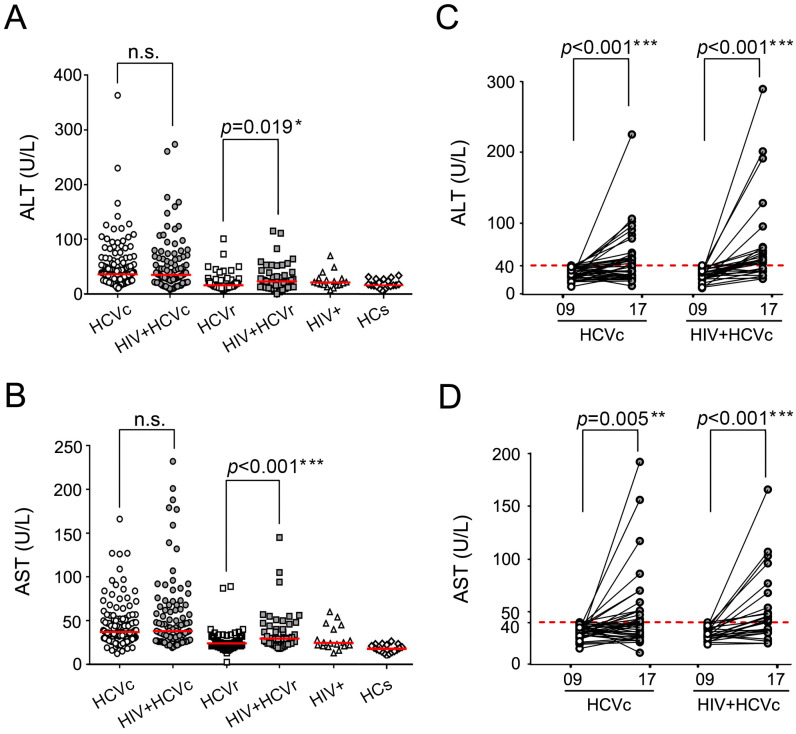
Comparison of ALT and AST levels and variations among different subgroups. (**A**) ALT and (**B**) AST levels in different subgroups at the first follow-up in 2009. The variation of (**C**) ALT and (**D**) AST levels from 2009 to 2017. To maintain neutrality, only individuals with normal test results in 2009 were compared for variance. The red solid line indicates the median levels. The red dashed line indicates normal values of reference. (*, *p* < 0.05; **, *p* < 0.01; ***, *p* < 0.001).

**Figure 3 viruses-14-01621-f003:**
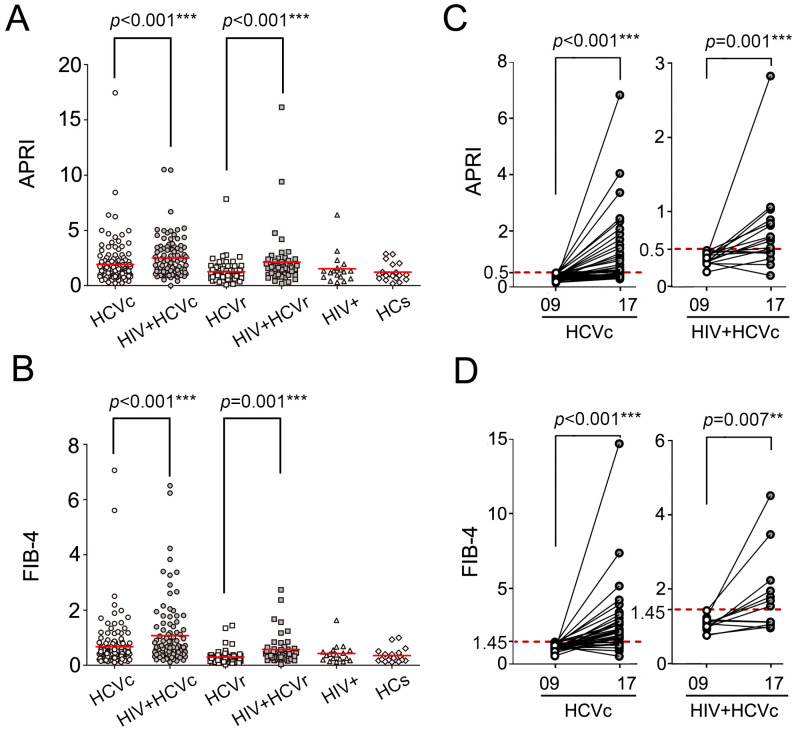
Comparison of APRI and FIB-4 levels and variations among different subgroups. (**A**) APRI and (**B**) FIB-4 levels in different subgroups at the first follow-up in 2009. The variation of (**C**) APRI and (**D**) FIB-4 levels from 2009 to 2017. To maintain neutrality, only individuals with normal test results in 2009 were compared for variance. The red solid line indicates the median levels. The red dashed line indicates normal values of reference. (*, *p* < 0.05; **, *p* < 0.01; ***, *p* < 0.001).

**Figure 4 viruses-14-01621-f004:**
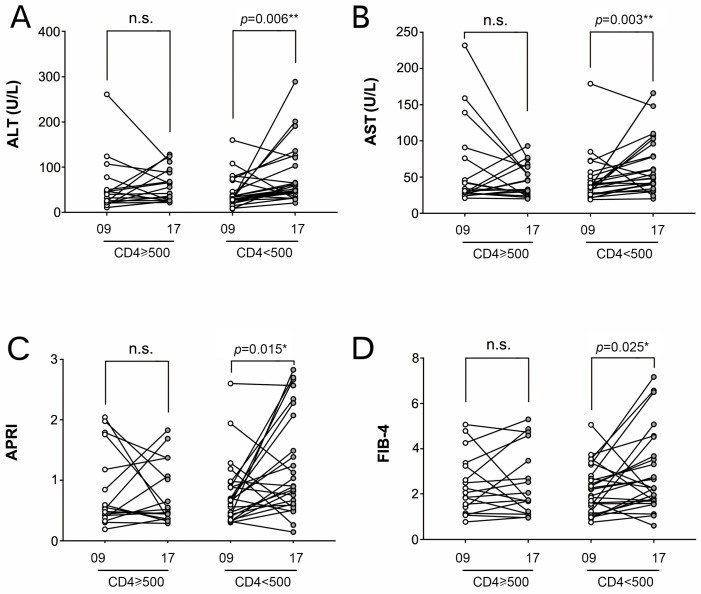
Comparison of variations of AST, ALT, APRI, and FIB-4 values between different CD4+ T cell subgroups from 2009 to 2017. (**A**) ALT, (**B**) AST, (**C**) APRI, and (**D**) FIB-4. (*, *p* < 0.05; **, *p* < 0.01; ***, *p* < 0.001).

**Table 1 viruses-14-01621-t001:** The demographic and clinical characteristics of all individuals enrolled in the study in 2009.

Variable	HCVc (*n* = 129)	HCVc/HIV (*n* = 98)	HCVr (*n* = 62)	HCVr/HIV (*n* = 48)	HIV (*n* = 18)	HCs (*n* = 18)
Age (y) ^1^	50.4 (12.8)	45.2 (9.8)	46.7 (12.7)	44.1 (9.2)	42 (13.1)	40.5 (13.5)
Gender (M/F)	61/68	45/53	13/49	20/28	8/10	4/14
Anti-HIV	Negative	Positive	Negative	Positive	Positive	Negative
HCV VL (log_10_ IU/mL) ^1^	6 (0.9)	6.2 (0.9)	None	None	None	None
Anti-HCV (S/CO) ^1^	14.4 (1.8)	13.2(3.2)	7.6 (4.1)	7.2 (3.5)	Negative	Negative
HCV genotype, n (%)						
1b	83	44	U.D.	U.D.	N.A.	N.A.
2a	46	54	U.D.	U.D.	N.A.	N.A.
Others	None	None	U.D.	U.D.	N.A.	N.A.
Biochemistry analyses ^1^						
ALT (IU/L)	49.7 (43.9)	51.5 (47.8)	21.3 (15.9)	30.5 (24.6)	24.8 (15)	21.1 (8.5)
AST (IU/L)	44.5 (24.7)	53.8 (42.5)	26.5 (13.2)	37.6 (23.5)	29.6 (13.5)	24.6 (6.8)
ALP (IU/L)	97.6 (50.3)	124.1 (59.7)	88.9 (34.9)	111.9 (58.4)	88.6 (31.2)	81.4 (33.5)
GGT (IU/L)	29.5 (26.7)	63.1 (73.8)	21.7 (27.3)	45 (45.9)	32.7 (20)	16.8 (6.8)
TBIL (μmol/L)	13.7 (3.5)	13.6 (5.2)	13.3 (3.4)	13.9 (3.4)	13.6 (4.9)	13.5 (3.2)
DBIL (μmol/L)	4.3 (1.3)	4.4 (1.7)	4.5 (2.4)	4.1 (1.4)	5.5 (1.6)	4.5 (1.5)
TP (g/L)	76.5 (7.8)	78.5 (6.7)	75.5 (6.1)	77.5 (5.2)	77.1 (8.4)	75.8 (2.7)
ALB (g/L)	44.2 (5.8)	44.6 (8.4)	42.8 (5.6)	44.6 (7.1)	44.8 (5.6)	39.9 (4.7)
CD4+T cell (cells/μL)	836 (315)	451 (238)	894 (339)	416 (233)	521 (344)	926 (379)
CD8+T cell (cells/μL)	653 (335)	994 (582)	697 (353)	1028 (378)	805 (263)	751 (357)
CD4+/CD8+ T cell	1.5 (0.7)	0.6 (0.5)	1.5 (0.7)	0.4 (0.2)	0.7 (0.5)	1.4 (0.6)

^1^ The data were presented as average and standard deviation; N.D.: undetectable; N.A.: not available; ALT: alanine aminotransferase; AST: aspartate aminotransferase; ALP: alkaline phosphatase; GGT: gamma-glutamyl transferase; TBIL: total bilirubin; DBIL: direct bilirubin; TP: total protein; ALB: albumin; HCVc: chronic hepatitis virus infection; HIV: human immunodeficiency virus; HCVr: spontaneous hepatitis C virus resolution.

**Table 2 viruses-14-01621-t002:** Mortality rates in HCV mono-infection and HCV/HIV coinfected patients from 2009 to 2017.

Variable	HCVc	HIV + HCVc	*p*-Value
Total patients in 2009 (n)	129	98	
Total death (n (% of total death))	16 (12.4%)	18 (18.4%)	0.213
AIDS-related death (n (% of total death))	0 (0%)	3 (16.6%)	0.087
Non-AIDS-related death (n (% of total death))	16 (100%)	15 (83.4%)	0.087
ESLD-related death (n (% of non-AIDS-related death))	5 (31.3%)	11 (73.3%)	0.009 **
Other cause of death (n (% of non-AIDS-related death)) ^1^	11 (68.7%)	4 (26.7%)	0.009 **

^1^ The 11 other causes of death in the HCVc group included the following: cerebral embolism (4), myocardial infarction (3), lung cancer (1), cerebral atrophy (1), gastric hemorrhage (1), and car accident (1); the four other causes of death in the HIV + HCVc group included the following: lung cancer (1), myocardial infarction (1), cerebral hemorrhage (1), and cerebral thrombosis (1). HCVc: chronic hepatitis C virus infection; HIV: human immunodeficiency virus; ESLD: end-stage liver disease. (*, *p* < 0.05; **, *p* < 0.01; ***, *p* < 0.001).

## Data Availability

The data used to support the findings in this study are available from the corresponding authors upon request.

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
