# Peer review of "Impact of HIV-1 Infection on the Natural Progress of an Anti-HCV Positive Population in an Impoverished Village in China from 2009 to 2017"

_viruses, 2022, doi:10.3390/v14081621_

Round 1
Reviewer 1 Report
Find attached the suggested changes in a PDF file

Author Response
Point-by-point responses to the comments from Reviewers
Reviewer(s)' Comments to Author:
In the paper entitled “Impact of HIV-1 infection on the natural progress of anti-HCV positive population in an impoverished village in China from 2009 to 2017” Li et colleagues describe an interesting longitudinal study about how HIV may increase the risk of liver damage in people living with HIV compared to HCV monoinfected subjects. However, some issues may limit the discussion and conclusions of this work in my point of view.
Major issues
1. ART-related liver toxicity
Those patients among the HIV arms received a standard ART based on nevirapine (NVP) and didanosine (ddI), stavudine (D4T), zidovudine (ZDV) or lamivudine (3TC). NVP, ddI and D4T have been appointed as frequent causes of liver toxicity within HIV infected patients, specially NVP. Thus, HIV infected patients did not also suffered damage from HCV coinfection. Mid and long-term exposure to these drugs may also be related to a more intense liver damage compared to HIV uninfected subjects. I think this one is a major confusion factor, that should be addressed in the paper. I consider useful to add this issue in the “limitations” section. In addition, I think it is necessary to address the relationship between ART and liver toxicity in the discussion.
A: Thank you very much for your helpful suggestions. Hepatotoxicity from antiretroviral medication is one of the primary concerns for HIV patients. We are also aware of this. Indeed, most of the patients with HIV mono-infection remained in a normal state without a significant rise in liver function indicators during the long-term follow-up, which may indicate to some extent that the hepatotoxic effects induced by antiretroviral therapy in this study were comparatively limited. However, it is possible that antiretroviral-induced hepatotoxicity was evident and a confounding factor at the initial follow-up in 2009 or earlier. To address this problem, we have added a note in the discussion's "limitations" section to clarify the study. Please see lines 307-313 of the revised manuscript for details.
2. Evaluation of liver fibrosis
Ultrasonographic elastometry has become a standard procedure to measure the liver fibrosis in HCV infected patients, combined with blood samples. However, no elastometry parameters have been included in the paper, and no grade of fibrosis is shown in the Table 1. Have been performed estastometry studies in the study population?
A: Thank you for your guidance. We need to illustrate that ultrasonographic elastometry was applied relatively late in China's undeveloped areas especially in the poor village where we conducted our survey. Therefore, in 2009, we were unable to perform ultrasonographic elastometry on patients included in the study. Fortunately, in the 2017 follow-up, we carried the relevant instruments and examined survivors for liver stiffness and grade, and we did not find significant differences when comparing these indicators between the HCV mono-infected and HIV+ HCV co-infected groups (data not shown). Table 1 does not provide this essential data because it only presented the baseline characteristics for the 2009 follow-up.
3. Line 248: “inappropriate sexual activities” is a controversial expression. What does it mean? Unprotected sex intercourse? I think this line should be reformulated.
A: Thank you very much for pointing this out. In the draft manuscript, we intended to refer to diseases caused by sexual routes such as men who have sex with men (MSM) (a group that had been the focus by other researchers). To prevent a controversial formulation, we have revised the original statement in the new manuscript. Please see lines 253-254 of the revised manuscript for details.
Minor
4. Only two HCV genotypes have been reported (1b and 2a). Could you check the original data? Genotypes 3 and 4 have not been detected in any subject?
A: Thank you very much for noticing this. We examined the raw data and confirmed that only two genotypes of HCV, 1b and 2a, were detected in this cohort of patients. Since HCV genotype distribution varies greatly by geography and population, HCV genotypes 1b and 2a are the most prevalent in China. In the northern region (including the village in this study), the genotypes were more homogeneous, with 1b and 2a predominating, whereas in the southern region, the HCV genotypes were more diverse, with 1b predominating and 2a, 3a, 3b and 6a each accounting for a larger proportion, for example, in Guangxi, Guangdong, China.
5. Total albumin and prothrombin ratio are also two good biomarkers for liver function. Could you add both biomarkers to the Table 1?
A: Thank you very much for your instructive advice. Total albumin and prothrombin ratio are indeed two indicators of liver function that cannot be ignored. Unfortunately, we lack data on prothrombin ratios due to experimental limitations. To make the data presentation more complete, we have added information on total protein (TP) and albumin (ALB) in Table 1 of the updated manuscript. Please refer to lines 202-208 and new Table 1 of the revised manuscript for details.
6. CD4/CD8 ratio is a useful biomarker among HIV-infected patients. A poor CD4/CD8 ratio is a predictor of a higher risk of AIDS and non-AIDS comorbidities. I think adding the CD4/CD8 ratio to the table 1 will be interesting.
A: Your suggestion is very pertinent. Considering the importance of CD4+/CD8+ T cell ratios in assessing disease progression of HIV patients, we have included CD4+/CD8+ ratios in HIV patients in the revised Table 1, as indicated in lines 202-208 and new Table 1 of the revised manuscript.
7. Mode of transmission for HIV should be added into the table 1.
A: In this study, more than 90% of the participants were former plasma donors (FPD) with a non-standard history of paid blood donation in the 1990s, and the remainder were their parents, spouses, or children. As a result, there is a high probability that the AIDS included in the study was caused by blood-borne transmission or sexual transmission. However, this is only a speculation and we cannot define the definitive source of the virus in this patient. Therefore, to avoid inappropriate statements, we did not list the modes of transmission of AIDS in the table.
8. Table 2 indicates the death rates among both cohorts. Other cause of death accounts nearly 69% of HCVc cohort. What are these main causes of death? Cardiovascular? Non-AIDS cancer?
A: Thank you for mentioning this. Because the majority of the patients followed were elderly, many other participants died from heart diseases, cerebral embolism, other cancers, or accidental causes during the nearly 10 years of follow-up. To clarify this point, we have added specific information on other causes of death in Table 2 of the new manuscript. Please refer to lines 241-246 and new Table 2 of the revised manuscript for details.
9. IL28B polymorphisms were widely employed to estimate the risk of failure among several anti HCV treatments. Indeed, certain IL28B are related to a lower risk of liver fibrosis progression. Was this biomarker explored in the study population?
A: Your suggestions are quite valuable. It is a pity that we were unable to examine IL28B in this study due to local experimental constraints. However, we appreciate your guidance and will consider further focusing and exploring this index in subsequent studies.

Reviewer 2 Report
Very important study, well provided. may be accepted in present form.
Author Response
We appreciate your positive and constructive comments and suggestions on our manuscript.
We have studied all comments carefully and tried our best to revise our manuscript according to the comments, which we hope will be met with approval. We believe the points emphasized by the reviewers are addressed in this revised manuscript.
Thank you again for your patience and sincere comments!
Round 2
Reviewer 1 Report
The author responses to my queries are excelent and the manuscript has improved in my view.